# Synthesis of Nanosilica for the Removal of Multicomponent Cd^2+^ and Cu^2+^ from Synthetic Water: An Experimental and Theoretical Study

**DOI:** 10.3390/molecules27217536

**Published:** 2022-11-03

**Authors:** Basel Al-Saida, Arwa Sandouqa, Reyad A. Shawabkeh, Ibnelwaleed Hussein

**Affiliations:** 1Department of Chemistry, Faculty of Science, Al-Balqa Applied University, Salt 19117, Jordan; 2Chemical Engineering Department, School of Engineering, The University of Jordan, Amman 11942, Jordan; 3Gas Processing Center, College of Engineering, Qatar University, Doha P.O. Box 2713, Qatar; 4Department of Chemical Engineering, College of Engineering, Qatar University, Doha P.O. Box 2713, Qatar

**Keywords:** nanosilica, adsorption, heavy metals, copper, cadmium, multicomponent adsorption

## Abstract

Copper and cadmium ions are among the top 120 hazardous chemicals listed by the Agency for Toxic Substances and Disease Registry (ATSDR) that can bind to organic and inorganic chemicals. Silica is one of the most abundant oxides that can limit the transport of these chemicals into water resources. Limited work has focused on assessing the applicability of nanosilica for the removal of multicomponent metal ions and studying their interaction on the surface of this adsorbent. Therefore, this study focuses on utilizing a nanosilica for the adsorption of Cd^2+^ and Cu^2+^ from water. Experimental work on the single- and multi-component adsorption of these ions was conducted and supported with theoretical interpretations. The nanosilica was characterized by its surface area, morphology, crystallinity, and functional groups. The BET surface area was 307.64 m^2^/g with a total pore volume of $4.95\times 10^{-3}$ cm^3^/g. The SEM showed an irregular amorphous shape with slits and cavities. Several Si–O–Si and hydroxyl groups were noticed on the surface of the silica. The single isotherm experiment showed that Cd^2+^ has a higher uptake (72.13 mg/g) than Cu^2+^ (29.28 mg/g). The multicomponent adsorption equilibrium shows an affinity for Cd^2+^ on the surface. This affinity decreases with increasing Cu^2+^ equilibrium concentration due to the higher isosteric heat from the interaction between Cd and the surface. The experimental data were modeled using isotherms for the single adsorption, with the Freundlich and the non-modified competitive Langmuir models showing the best fit. The molecular dynamics simulations support the experimental data where Cd^2+^ shows a multilayer surface coverage. This study provides insight into utilizing nanosilica for removing heavy metals from water.

## 1. Introduction

Water pollution by dyes and toxic heavy metals represents a principal global environmental issue [1,2,3,4]. Heavy metals, such as Cu(II), Cd(II), Cr(VI), Cu(II), Hg(II), and As(III, V), are some of the most harmful pollutants due to their high toxicity, solubility in water and impact on aquatic ecosystems, and hazardous environmental effects [5,6,7]. Unlike organic pollutants, heavy metals are non-biodegradable and accumulate in living organisms. There are many sources that discharge heavy metals into water systems, such as metal plating, smelt, and electrolysis industries [8,9]. Releasing large amounts of heavy metals into water systems has severe environmental impacts. Copper and cadmium, for example, are considered among the top toxic substances that need urgent treatment [10]. According to the recommendations of the World Health Organization (WHO), drinking water’s standards for Cd^2+^ and Cu^2+^ levels are 0.003 ppm and 1.3 mg/L, respectively [11]. The United States Environmental Protection Agency (USEPA) has set these values in drinking water to 0.005 ppm and 2.0 mg/L, respectively [12,13].

Numerous treatment methods have been adopted to remove these heavy metals from down streams using precipitation [14], ion exchange [15], photo-Fenton oxidation [16], flocculation [17], electrochemical treatment [18], and adsorption [3,19,20]. Some of these methods have limitations, including producing harmful byproducts [21], high energy consumption, and high installation costs [22]. Adsorption, on the other hand, has gained great interest from many researchers due to its low cost of operation, ease of handling, high efficiency at low concentrations, and desorption ability [23,24].

Many adsorbents are available in the literature for removing heavy metals from aqueous solutions. These include activated lignocellulosic materials, alumina, silica, aluminosilicates, and nanodiamond materials [25,26,27,28,29].

When dealing with the fate of these heavy metals in water aquifers, silica (SiO_2_) is the most abundant natural adsorbent for capturing these metals before reaching the water blanket. It has gained significant interest because of its unique properties as an eco-friendly adsorbent with very low toxicity and rigidity [30,31]. Due to its high surface area (ca. 180–400 m^2^/g) [32], a wide range of zero point of charge (depending on impurities pH_ZPC_ 3.8–7.1) [33], stability, strength, and recyclability, nanosized silica has been used in many applications [34]. However, the literature has little information on using this adsorbent for the multi-component removal of copper and cadmium.

This research aims to prepare a nanoscale silica using the sol–gel method and apply this material for the simultaneous removal of Cu^2+^ and Cd^2+^ from water. Adsorption isotherm experiments using a multi-component system coupled with isotherm models are presented.

## 2. Theoretical

The Langmuir isotherm is used to predict the monolayer coverage of single metal ions on the surface of the produced nanosilica. The model assumes a finite and uniform number of adsorbed sites, where the surface reaches a maximum adsorption capacity at equilibrium. This model is presented in Equation (1) [35]:(1)$$q_{e}=\frac{QkC_{e}}{1+kC_{e}}\;$$
where $q_{e}$ [mg/g] is the solid uptake at a given equilibrium concentration of a metal ion in solution, $C_{e}$ [mg/L]; $Q_{max}$ is the maximum metal ion uptake by the surface [mg/g]; and k is a constant related to the affinity of the binding sites and is directly proportional to the energy of adsorption (L/mg).

To predict the surface energy heterogeneity of the active adsorption sites, the Freundlich isotherm is employed using the following equation [36]:(2)$$q_{e}=k_{L}\;Ce^{\propto }$$
where $k_{L}$ and ∝ are the isotherm constants.

The Dubinin–Radushevich (DR) isotherm is applied to distinguish the type of adsorption of the metal ions based on the mean free energy of interaction and pore-filling, instead of layer-by-layer coverage [37]. The amount of ion uptake, $q_{e}$ [mg/g], is related to the adsorption potential, ε [kJ/mol], based on the following equations:(3)$$q_{e}=Q\;e^{-k_{d}\;\varepsilon^{2}}\;$$
$$\varepsilon =RTln\left(1+\frac{1}{Ce}\right)\;$$
where $Q_{max}$ is the maximum metal ion uptake by the surface [mg/g], and $k_{d}$ [${\mathrm{mol}}^{2}/{\mathrm{kJ}}^{2}]$ is a constant related to the mean adsorption energy, E [kJ/mol], as calculated using the following equation:$$k_{d}=\frac{-1}{2E}$$

## 3. Material and Methods

### 3.1. Materials and Reagents

Tetraethoxysilane (TEOS 98% Fluka), nitric acid (AR 65–68% JHD), sodium hydroxide (99% GCC), and ethanol absolute (AR 99.9% Merck) were used.

### 3.2. Synthesis of the Nanosilica Material

Tetraethoxysilane (TEOS) is a precursor for synthesizing nanosilica using the sol–gel method (Figure 1). First, a mixture (A) of 10 mL of ethanol, 10 mL of distilled water, and 5 mL of concentrated HNO_3_ was added to a mixture (B) of 20 mL of ethanol and 20 mL of tetraethoxysilane using dropwise addition with continuous stirring for 15 min. The mixture was stirred for 15 min and left heated at around 80–90 °C for one minute; the milky formed gel indicated the formation of nanoparticles. Finally, the nanoparticles were aged for 24 h, dried at 50 °C, and then calcined for two hours at 400 °C to obtain the nanosilica (NS) [38].

### 3.3. Characterization of Nanosilica Materials

The synthesized nanosilica was fully characterized using a Fourier transform infrared (FTIR) spectroscopy, an X-ray diffraction (XRD), a differential scanning calorimetry (DSC), a scanning electron microscopy (SEM), and an atomic absorption spectroscopy (AAS).

### 3.4. Adsorption Isotherms

The adsorption isotherm was determined by mixing 0.1 gm of the adsorbent with 50 mL of a metal solution that included varying initial metal concentrations ranging from 25–200 ppm. The pH of the multicomponent solution was adjusted to 6.0 using NaOH (1 M) and kept at 25 ± 1 °C (room temperature) for 72 h to attain equilibrium. Then, the concentrations of Cu^+2^ and Cd^+2^ were measured using an atomic absorption spectroscopy.

## 4. Results and Discussion

### 4.1. Structure Characterization of the Nanosilica

The XRD chromatogram for the produced nanosilica (Figure 1) shows a broad peak at ca. 23 2θ with a peak width of 20 2θ and an intensity of 850 counts. This peak broadness represents an amorphous silica due to non-arranged crystals in 3D space; hence, X-rays are scattered in several directions [39,40]. The Scherrer equation was used to determine the crystal size, $d\;\left(\mathrm{nm}\right)$, as follows:(4)$$d=\frac{K\;\lambda }{B\;\cos\left(\theta \right)}$$

The shape factor of the measured crystals, K, is approximated to 0.94 for the Full Width at Half the Maximum peak of spherical crystals with cubic symmetry. The wavelength, λ, has a value of 0.15418 (Cu K-alpha). The value of B is 9.6 2θ, considered at the FWHM. Consequently, the grain size of the silica is calculated to be 3.6 nm, which agrees with published work [41,42]. The broad diffracted peak of the produced material is attributed to the amorphous structure of the sample.

Differential scanning calorimetry for the produced silica sample was conducted in the temperature range from 0 to 300 °C (Figure 2). At a temperature of around 33.5 °C, the silica particles show removal of nonbounded free water that has a weak interaction with the surface [43], resulting in the endothermic effect of water dropping from about 0 to −1.25 mW. Then, there is a gradual increase in heat flow with increasing temperature. In addition, a continuous weight loss appears with increasing temperature up to 104–110 °C with a change in slope at 73 °C. At a temperature of ca. 243 °C, there is a change in the slope that might be attributed to structural hydroxyl elimination, according to Figure 2:

The silica nanoparticles’ functional groups before interaction with the cadmium and copper ions were investigated, and the FTIR is shown in Figure 3. An absorption peak at 3160 cm^−1^ and at 1636 cm^−1^ is attributed to the -OH group and H–O–H vibrations [44]. The peak belonging to the Si–O–Si stretching vibration is at 1041 cm^−1^, and the peak around 778 cm^−1^ corresponds to the Si–O–Si symmetric stretching [45].

The BET adsorption isotherm of N_2_ at 77 K by the produced silica sample is illustrated in Figure 4. It is shown that, with an increase in the relative pressure of nitrogen from 0.05 to 0.3, the quantity of nitrogen uptake increases linearly from 65 to 98 cm^3^/g at STP with a slope of 133.3. A plot of the linearized BET isotherm yields a monolayer coverage of 70.67 cm^3^/g and a constant value of 140. This produces a single-point surface area of 300.4 m^3^/g and a multi-point value of 307.6 m^2^/g, as shown in Table 1. The sample surface area obtained from the Langmuir isotherm offers a value of 461 m^3^/g. A micropore volume of 0.0049 cm^3^/g and an area of 12.89 m^2^/g are achieved at a relative pressure (P/Po) of less than 0.1. Hence, the micropore depth is 0.384 nm. The total pore radius is obtained using the Kelvin Equation (Equation (8)) and shows a value of 49 nm to indicate a mesoporous structure of the sample.

(5)$$r_{m}=\frac{-2\;\sigma \;v_{m}}{RT\;ln\left(\frac{p}{p_{o}}\right)}\;$$
where σ is the surface tension of nitrogen (8.85 erg/cm^2^); $v_{m}$ is the monolayer coverage obtained at the relative pressure value, $p/p_{o}$ of 0.3; R is the ideal gas constant ($8.314\times 10^{7}$ erg/K.mol); and *T* is the temperature (77 K). The adsorbed layer thickness and porosity data were estimated using the Harkins–Jura thickness equation:(6)$$t={\left[\frac{0.1399}{0.034-log\left(\frac{p}{p_{o}}\right)}\right]}^{1/2}\;$$
where t is the thickness of the BET multilayer of adsorbed N_2_ molecules at a relative pressure $p/p_{o}.$ Figure 4 also illustrates the variation in the N_2_ adsorbed with the adsorption layer thickness. The linear range of the t-plot lies within the adsorbed layer thickness with values of 0.37–0.48 nm.

The SEM image of the produced nanosilica is presented in Figure 5. It shows irregular amorphous shapes ranging from 30 to 300 nm; the surface morphology is non-uniform with a pore diameter of sub-meso size. The snowy-shaped agglomerated particles show micropores in the range of sub-nanometer. Some slits and cavities appear between some tiny silica crystals in between the agglomerated particles.

### 4.2. Adsorption Isotherms of Cu^2+^ and Cd^2+^ by the Nanosilica

The adsorption equilibrium for both copper and cadmium ions is illustrated in Figure 6. Both ions’ uptake shows a gradual increase with a multilayer of adsorption, which can be attributed to the surface heterogeneity of the produced nanosilica [46]. Alan et al. showed that the surface charge of nanosilica is directly affected by the cavities within the silica structure, where it increases with increasing particle roughness and curvature hills of the surface structure [47]. Cárdenas and Müller used molecular dynamics simulation to study the Lennard–Jones fluid’s behavior within the nanopores of different shapes. They proved that different pore shapes, sizes, and cavities have various adsorption capacities, where capillary condensation occurs in the acute corners of the surface [48]. Alosaimi et al. showed that Cd^2+^ is coordinated with the carboxylate group on the surface of the silica composite to form multilayer adsorption [49]. Similar results for multilayer adsorption of Cu^2+^ by silica nanoparticles from leaf biomass have also been reported [50]. The Langmuir, Freundlich, and Dubinin–Radushevich isotherm models were used to fit the experimental data, and the values of these models’ parameters are listed in Table 2. The Freundlich model best fits these experimental data, with regression coefficients of 0.974 and 0.990 for both Cu^2+^ and Cd^2+^ isotherms (Figure 7a,b). At the same time, the saturation capacities for both solutes obtained by the Langmuir model are 29.28 and 72.13 mg/g, respectively. When comparing to other published work, Rauf Foroutan et al. used a nanosilica from white sandstone for the removal of cadmium ions; the results showed that the maximum uptake of Cd(II) adsorbed only 55.13 mg/g [30]. Mahmoud et al. used SiO_2_ nano-powder to study Cd(II) biosorption from aqueous solutions; the maximum cadmium capacity determined was 600 μmol/g (67.2 mg/g) [51].

### 4.3. Binary Component Isotherm

Binary isotherm experiments were carried out using 80 samples; each had the same adsorbent concentration of 100 mg/50 mL solution with a temperature of 22 ± 1 °C. All multicomponent adsorption isotherms were measured at equilibrium concentrations (0 for the pure component to 120 mg ion/L for Cu^2+^ and Cd^2+^). Figure 8a,b illustrate the effect of the equilibrium concentration of both copper and cadmium ions on their uptake on the nanosilica, respectively. In both figures, the increase in each metal’s concentration results in a decrease in the reduction efficiency of the adsorbent. It is also noticed that Cd^2+^ has higher uptake than that of Cu^2+^ as a result of its higher affinity toward the adsorbent surface. Cadmium ions compete for the active surface of the adsorbent more than copper ions, which results in a decrease in Cu^2+^ uptake with increasing Cd^2+^ in the solution. A maximum adsorption capacity of 18 mg/g of pure copper is obtained at an equilibrium concentration of 120 mg/L.

Similarly, cadmium ion uptake obtains a value of 23 mg/g at these given concentrations. Increasing the concentration of cadmium ions in the solution decreases copper uptake to 12 mg/g when 120 mg/L of cadmium is presented in the solution. This decrease in uptake follows a linear relation when adding cadmium ions into the solution. On the other hand, the maximum adsorption capacity of pure cadmium ions obtained at an equilibrium concentration of 120 mg/L is 23 mg/g. This value drops to 11 mg/g when the copper concentration in the solution is 120 mg/L.

This antagonism effect is related to the competitive adsorption on the fixed surface area of the adsorbent. When both ions compete at a concentration of 120 mg/L, a reduction in the adsorption capacity of 33% and 52% is obtained for Cu^2+^ and Cd^2+^, respectively. It is also noted that copper ions compete on the surface of adsorbent at a higher order of magnitude than cadmium. This is related to the physicochemical properties of each ion, where the relative binding strength and the Pauling electronegativity of Cu^2+^ are higher than those of Cd^2+^ [52]. The difference in electronegativity between the oxygen atoms on the surface of the silica and cadmium is higher than that of copper. Therefore, the adsorption of cadmium ions on the surface is favored over copper. The addition of copper ions to the cadmium solution decreases the electronegativity difference, thereby decreasing surface uptake.

These results were compared to other published results regarding competitive adsorption on different types of modified silica (Table 3): the total uptake of these ions by the developed adsorbent is higher than that obtained when magnetite was coated on the silica surface and is comparable to that obtained for nitrogen-doped nanosilica.

### 4.4. Regeneration of the Adsorbent

Nanosilica samples were regenerated using 50 mL of 1 M hydrochloric acid solution. Then the samples were washed with distilled water several times until the solution pH reached a neutral value (pH 6.5–7). Table 4 illustrates the adsorption capacity of the adsorbent with the number of regeneration cycles. Isotherm experiments showed a minor reduction in the metal ions’ capacities, with an average decrease in adsorbent activity of 8% being obtained at the end of the fifth cycle.

### 4.5. The Molecular Dynamics of Cu and Cd on SiO_2_ Crystalline Structure

Molecular dynamics simulations were carried out using Accelrys Material Studio (V7). A semi-hydrated silica crystal with dimensions of (1 nm × 1 nm × 1.5 nm) was developed as an adsorbent to be used for the theoretical adsorption calculation of Cu^2+^ and Cd^2+^ from an aqueous solution. A cleaved plane (0 1 0) with a fractional thickness of one shows 24 silicone and 48 oxygen atoms. The total charge of the structure was adjusted based on its isoelectric charge obtained at pH 6 to include 2000 effective charges in a 1 µm^2^ silica sphere [56].

Ions of copper and cadmium were built using 3D atomistic drawing, and the charges for both metals were adjusted to be (+2) each and loaded equally on the surface of the silica. Figure 9 shows the unit cell for the hydrated silica, and the copper and cadmium ions as cleaned to maintain minimum free energy of interaction between the atoms. The Forcite module, with the COMPASS Forcefield and the Ewald electrostatic summation method, was used to optimize the surface geometry of the silica with an accuracy of 0.0001 kcal/mol. Both Cu^2+^ and Cd^2+^ were also optimized similarly.

An adsorption locator for both Cu^2+^ and Cd^2+^ on the silica surface was performed using the universal forcefield and the Ewald summation method, with 100,000 loading steps and one heating cycle. The interaction energy was measured at 11,000 steps, with an incremental 1000 steps each. The framework charge contains 48 electrons/cell with a maximum of five ions for each Cu^2+^ and Cd^2+^. The loadings of both ions are shown in Figure 10, where both ions demonstrate homogeneous uptake within the silica structure. The interaction energy during the adsorption rate is related to the van-der-Waals energy and the electrostatic and intermolecular energies, as shown in Figure 11. The van-der-Waals and intermolecular energies demonstrate a minor change, while the electrostatic energy decreases with the loading steps. The isosteric heat of adsorption for the interaction of both ions with the surface shows an increase in uptake (Figure 12). At the same time, the energy of interaction of the cadmium ions with the surface is higher than that of the copper ions, which agrees with the obtained experimental results. The gradient of the isosteric heat of adsorption decreases with loading, which is attributed to the loss of free surface energy needed for further ion uptake. Moreover, both ions are loaded equally on the surface at the same energy range (400–1200 kcal/mol), but cadmium ions are further adsorbed at a higher energy range (1500–2400 kcal/mol). These findings support the higher cadmium uptake by the surface where a multilayer of adsorption occurs.

Theoretical equilibrium uptake of both ions was conducted using the Sorption module. The Metropolis and Monte Carlo method has 219 million random seeds and 10,000 equilibrium steps. The universal forcefield energy was employed correctly with the Ewald electrostatic summation method. The isotherm results were obtained as the average loading of molecules per unit cell. Figure 13 and Figure 14 illustrate the average loading for each ion in a multi-component system, where cadmium ions show better uptake than copper ions. The maximum loadings for these ions per unit cell are 69 and 31, where each ion’s Langmuir equilibrium trend is noticed.

## 5. Conclusions

Naturally occurring and synthesized silica are among the effective adsorbents used for heavy metal removal from water. A nanosilica was synthesized using the sol–gel technique and characterized for its surface characteristics and adsorption competency. An irregular amorphous shape with several cavities and slits, a total surface area of 307.64 m^2^/g, and a total pore volume of $4.95\times 10^{-3}$ cm^3^/g was obtained.

The competitive uptake of heavy metals, such as Cd^2+^ and Cu^2+^, on the silica surface determines the adsorption capacity of these ions. Single component systems for the removal of Cd^2+^ and Cu^2+^ from water were investigated using experimental and theoretical techniques with the produced nanosilica. The results showed that cadmium ions are adsorbed electrostatically by the nanosilica surface due to the high energy of interaction, which forms a multilayer of uptake, resulting in a maximum single adsorption capacity of 72.13 mg/g. Copper adsorption shows an uptake capacity of 29.28 mg/g. These values are reduced by 52% and 33% when competitive adsorption occurs at a maximum equilibrium concentration of 120 mg/L for each ion.

Molecular dynamic simulations of the interaction of Cd^2+^ and Cu^2+^ on the modeled silica surface demonstrated that the isosteric heat of adsorption is a significant factor in the competitive uptake of these ions. This interaction energy between the Cd^2+^ ions and the silica is higher than that of the Cu^2+^ ions; therefore, a multilayer coverage could be obtained for the Cd^2+^ ions. The nanosilica demonstrates the feasibility of removing Cd^2+^ and Cu^2+^ ions due to the difference in electrostatic charge.

## Data Availability

The data presented in this study are available on request from the corresponding author. The data are not publicly available due to the limitation of the university server.

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
