# Peer review of "Synthesis of Nanosilica for the Removal of Multicomponent Cd2+ and Cu2+ from Synthetic Water: An Experimental and Theoretical Study"

_molecules, 2022, doi:10.3390/molecules27217536_

Round 1
Reviewer 1 Report
Review of an Artilce:
«Synthesis of nanosilica for the Removal of Multicomponent Cd2+ and Cu2+ from Wastewater: Experimental and Theoretical Study»
by Basel Al-Saida, Arwa A. Sandouqa, Reyad A Shawabkeh, Ibnelwaleed A. Hussein
In Molecules (ISSN 1420-3049).
Round 1
This work demonstrated the possibilities of the as-synthesized nanosilica for the removal of multicomponent Cd2+ and Cu2+ from wastewater.
The article was presented in a well-structured manner. Unfortunately, several statements within have weak evidence that should be expanded with more valuable proof. The novelty of the work raises several questions. Therefore, the referee suggested that the manuscript should be considerably improved in a major revision. The following is a list of specific concerns.
Introduction.
1. The purpose of the Introduction is to attract the reader of Molecules by what he can find here in contrast to other similar publications related to silica.
- As well, please, use in the introduction section the following recent important research:
· https://www.sciencedirect.com/science/article/pii/S0925963520306749 (for comparison with other non-metal materials)
· https://www.sciencedirect.com/science/article/pii/S1383586621000435 (recent advances for effective transition metal removal by magnetic mesoporous silica)
etc.
- The pros and cons of sol-gel synthesis of silica should be discussed and reviewed.
Material and Methods
2. Line 113. 1N solution of NaOH
- IUPAC and NIST discourage the use of normality. Should be recalculated and used as a 1M solution of NaOH.
3. Then, the Concentration of Cu+2 and Cu+2 was measured by Atomic Absorption Spectroscopy.
- Presumably, Cu+2 and Cd+2, isn’t it?
- As well, the condition and all specifications of the used equipment should be presented. Is it Flame or Furnace AAS?
- Specify, the Dilution ratio of all samples.
- Provide any data of accuracy (trueness and precision) assessment for Cu and Cd measurements according to ISO 5725 or any other documents.
4. Line 131. Differential scanning calorimetry experiments should specify any scheme of reactions.
5. Is it possible of using pristine nanosilica repeatedly (regeneration)? How many cycles is it up to?
6. Extra experiments should be done.
- The estimation of maximal concentration should be increased up to 1 g/l (or 10g/L) to prove the maximal capacity of synthesized sorbent. Because of (fig.7) equilibrium concentration up to 200 ppm is not representative. The steady-state (plateau-level should be approached) if possible.
7. Any comments concerning interfering influence other ions Pb(II), Mn(II) etc. for real sample sorption should be provided.
8. Reference list should be eapanded with relevant work of 2022.
Style guide issues
Line 58: ca. should be italic ca.;
English spelling should be double-checked.
Reviewer 2 Report
Comments
In this work Al-Saida et al., reported the synthesis of nanosilica for the removal of multicomponent Cd2+ and Cu2+ from wastewater. In my opinion, the article is not ready to by published in the present version. Major Revisions for reconsideration.
The following points need to be clarified or addressed to improve the quality of the manuscript:
1. The organization of the manuscript should be checked, especially the serial number of the headlines should be corrected.
2. References should be cited to support the Langmuir and Freundlich adsorption isotherms. Few important references related to Langmuir and Freundlich adsorption isotherms (e.g., Crucial roles of graphene oxide in preparing alginate/nanofibrillated cellulose double network composites hydrogels, Chemosphere, 2021,263: 128240; Graphene oxide reinforced alginate/PVA double network hydrogels for efficient dye removal. Polymers. 2018, 10(8), 835) are suggested to be cited to see the developments.
3. There are some problems for your description of adsorption experiments. When the author describes the adsorption experiments, the author should emphasis the condition of the adsorption, such as initial concentration of ions. The author should better label the information in the annotation of Fig.7, Fig.8, and Fig. 9 and so on.
4. The abstract can be improved, and the creative point and the advantages of your method should be mentioned.
5. It will be better to include a table comparing the SiO2 nanomaterials for their application for Cu and Cd ions removal, in order to explore superiority of the present results.
Round 2
Reviewer 1 Report
Review of an Artilce:
«Synthesis of nanosilica for the Removal of Multicomponent Cd2+ and Cu2+ from Wastewater: Experimental and Theoretical Study»
by Basel Al-Saida, Arwa A. Sandouqa, Reyad A Shawabkeh, Ibnelwaleed A. Hussein
In Molecules (ISSN 1420-3049).
Round 2
The authors have done sufficient work to improve the quality of the text. All critical deficiencies of the previous version have now been revised. I hope different experiments will be done in future articles (increasing Equilibrium Concentration up to g/L).
I think minor changes should be made to the polishing of the manuscript.
The following is a list of specific concerns.
1. Still, in work, many figures should be combined or transferred to the Supplementary material section.
2. “Scheme 2. Dehydroxylation of silica at high temperature” have a lack of information. I think the scheme of reaction should be depicted. For example, 2>Si–(OH)2 = >Si–O–Si< + H2O; or with other ways. There are a lot of non-discussed peaks on the DSC of nano-silica particles.
3. Figures 4, 5, 7, 8, and 9 and table 2 need to be reconsidered. The confidence intervals should be added.
I think it is necessary to accept the work after minor revision.
Reviewer 2 Report
The authors have answered all my questions. The reviewer suggests the acceptance of the paper.
Author Response
No comments from the reviewer. Thanks for recommending acceptance.